# Trends and Disparities in Liver Transplantation in the United States: A Nationwide Analysis of Demographic, Clinical, and Socioeconomic Factors (2016–2021)

**DOI:** 10.3390/medsci13020066

**Published:** 2025-06-01

**Authors:** Vignesh Krishnan Nagesh, Vivek Joseph Varughese, Marina Basta, Emelyn Martinez, Shruthi Badam, Lokaesh Subramani Shobana, Abdifitah Mohamed, Alin J, Simcha Weissman, Adam Atoot

**Affiliations:** 1Department of Internal Medicine, Hackensack Palisades Medical Center, North Bergen, NJ 07047, USA; msambasta@gmail.com (M.B.); emelyn03.em@gmail.com (E.M.); b.shruthi01@gmail.com (S.B.); lokaesh0904@gmail.com (L.S.S.); simchaweissman@gmail.com (S.W.); adamatoot.md@gmail.com (A.A.); 2Department of Internal Medicine, University of South Carolina, Columbia, SC 29208, USA; 3Department of Internal Medicine, University of Washington, Seattle, WA 98195, USA; drsharif@uw.edu; 4Department of Internal Medicine, Medical College Thiruvananthapuram, Thiruvananthapuram 695011, Kerala, India; alinj71@gmail.com

**Keywords:** liver transplantation, racial disparities, healthcare access, in-hospital mortality, national inpatient sample (NIS)

## Abstract

Background: Liver transplantation has become the standard of care for patients with end-stage liver disease. Despite advances in surgical techniques, immunosuppression, and perioperative care, disparities in access and outcomes persist across demographic and socioeconomic lines. Objective: To assess trends and disparities in liver transplant admissions in the United States from 2016 to 2021, examining demographic patterns, in-hospital mortality, hospital charges, length of stay, and socioeconomic factors. Methods: Using the National Inpatient Sample (NIS) from 2016 to 2021, we identified liver transplant admissions using ICD-10 PCS codes 0FY00Z1 and 0FY00Z2. Demographic characteristics (age, sex, race, insurance status, and income quartile), clinical outcomes, and resource utilization metrics were analyzed. One-way ANOVA and Hensel’s test were used to assess variance and distribution homogeneity, with a significance threshold of *p* < 0.05. Results: A total of 9677 liver transplant admissions were analyzed. The mean recipient age remained stable (51–52 years), with males comprising ~62% of transplants. White patients constituted the largest group of recipients (~66–68%), followed by Hispanic (~14–17%) and Black patients (~7–10%). The proportion of transplants relative to liver failure admissions remained stable across racial groups, indicating no widening racial gap during the study period. In-hospital mortality post-transplant remained low (2.37–3.52%) and did not differ significantly by race (*p* = 0.23), sex (*p* = 0.24), or income quartile (*p* = 0.13). Similarly, Charlson Comorbidity Index > 5 did not predict inpatient mortality (*p* = 0.154). Hospital charges ranged from $578,000 to $766,000, with an average stay of ~21 days. Conclusions: Liver transplantation outcomes, including in-hospital mortality, appear consistent across demographic and socioeconomic groups once patients are admitted for transplant. However, broader disparities in access persist, necessitating further research into pre-transplant barriers and long-term outcomes. These findings support the need for equitable healthcare strategies aimed at optimizing transplant candidacy and survival across all populations.

## 1. Introduction

Liver transplantation has emerged as a definitive treatment for patients with end-stage liver disease, offering hope where medical management alone often proves insufficient [1]. Over the last few decades, surgical techniques, immunosuppressive regimens, and perioperative care have significantly improved, leading to expanded the pool of transplant candidates and improved long-term survival rates [2]. Nevertheless, the global burden of chronic liver disease continues to rise, fueled by factors such as non-alcoholic fatty liver disease (NAFLD), chronic viral hepatitis, and alcohol-related liver damage [3,4]. Consequently, the demand for donor livers remains persistently high, leading to critical shortages and the need for equitable allocation strategies [5].

In addition, disparities in accessing liver transplantation persist across various demographic groups despite extreme efforts to make liver transplant accessible geographically. Socioeconomic status, race, and ethnicity can all shape a patient’s likelihood of receiving timely evaluation and transplantation [6,7]. Furthermore, co-morbid conditions—particularly those captured by indices such as the Charlson Comorbidity Index—can influence surgical candidacy, post-transplant complications, and mortality [8]. As a growing number of healthcare systems strive to provide universal or expanded coverage, insurance status has also surfaced as a determinant of both candidacy and quality of postoperative care [9]. Disentangling the impact of these factors on survival and resource utilization is critical to ensuring that liver transplant programs continue to evolve in an inclusive and patient-centered manner.

In the following analysis, we examine nationwide data collected from 2016 to 2021 on liver transplant admissions to assess key patterns and outcomes, including overall mortality, hospital charges, length of stay, and demographic distributions. By focusing on these clinical and socioeconomic variables, our study aims to offer a focused view of ongoing challenges in liver transplant care. Ultimately, these findings can help inform targeted interventions and policies that advance equity, optimize resource use, and improve patient outcomes in this high-stakes clinical setting.

## 2. Methods

National Inpatient Sample (NIS) for the years 2016 to 2021 was used for the analysis, and STATA 18.5 MP (StataCorp, College Station, TX, USA) was used to do the statistical analysis. ICD 10 PCS codes 0FY00Z1, 0FY00Z2 were used to select admissions that underwent liver transplantation. Mean age, sex, race, and primary co-payer were stratified for the admissions and trends analyzed across the years. The total number of admissions for liver failure (K729*) was selected and compared with liver transplantations for the corresponding year. Population stratification was conducted for the total number of liver failure admissions and admissions that underwent liver transplantations. Age, sex, race, and median household income were stratified and analyzed in both groups. The analysis was non-weighted given the sample size. Homogeneity of distribution was tested using Hensel’s test, and one-way ANOVA was used to study variance of patient factors in the liver failure and the liver transplant group. A two-tailed *p*-value < 0.05 was used to determine statistical significance.

## 3. Results

The demographics of patients analyzed for this study from the years 2016–2021, along with the mean of total hospital charges and mean length of stay, have been summarized in Table 1.

Between 2016 and 2021, the total number of liver transplants in the U.S steadily increased from 1437 in 2016 to 1782 in 2021. The mean age of recipients remained relatively stable, averaging around 52 years. Males consistently accounted for the majority of transplant recipients, ranging from 61% to 64% each year. White patients represented the largest racial group undergoing transplantation, making up around two-thirds of cases annually. Black and Hispanic patients accounted for 7–9% and 14–17%, respectively. In-hospital all-cause mortality stayed relatively low, fluctuating between 2.37% and 3.52%. Hospital charges rose over time, with mean total charges increasing from approximately $578,000 in 2016 to over $766,000 in 2021. The average length of stay remained stable, typically around 20–22 days. Regarding insurance coverage, private insurance was the most common primary payer throughout the period, covering 47–52% of patients. Medicare and Medicaid followed, with Medicare covering about 25–31% and Medicaid covering 15–18% of transplants annually.

From 2016 to 2021, hospital admissions for hepatic failure have risen significantly, outpacing the increase in liver transplants. In Figure 1, admissions increased from just over 3000 cases in 2016 to nearly 5800 by 2021—a clear indicator of growing liver disease burden. While liver transplants also showed a modest upward trend during this period (rising from ~1450 to ~1780), the gap between need and availability continues to widen. This underscores a critical challenge in transplant medicine: the persistent shortage of organs compared to the increasing demand.

In-hospital mortality has shown a fluctuating but overall rising trend from 2016 to 2021.

As shown in Figure 2, the number of deaths increased sharply from 34 in 2016 to a peak of 54 in 2017, followed by a slight dip in 2018. However, mortality steadily rose again in subsequent years, reaching the highest point of 55 in 2021. This upward trajectory, particularly in the context of increasing hepatic failure admissions, may reflect worsening disease severity, limited transplant availability, or systemic healthcare challenges.

Figure 3 tracks two metrics from 2016 to 2021: male count, and female count. The male count starts at approximately 950 in 2016 and steadily increases till 2021, indicating a slight upward trend. The female count, be-gins at around 600 in 2016, dips slightly in 2018, and then gradually rises to 689 by 2021, showing a modest increase over the years. Overall, the male count con-sistently exceeds the female count, with both showing gradual growth, while the mean age remains stable.

Figure 4, illustrates the mean length of stay. The mean hospital charges start at around 600,000 in 2016, rise slightly to 650,000 in 2018, dip to approximately 600,000 in 2019, and then increase steadily to about 750,000 by 2021, reflecting a general upward trend with a temporary decline. Overall, hospital charges exhibit variability and growth.

Figure 5 observe the number of patients receiving liver transplants categorized by their primary insurance provider over the six-year period. Medicare and Medicaid consistently covered a smaller number of transplant recipients compared to private insurance each year. While the number of transplants covered by Medicare and Medicaid showed some fluctuation between 2016 and 2021, the number of liver transplants with private insurance as the primary co-payer generally trended upward, showing a notable increase, particularly in the final year of 2021.

As shown in Table 2, over the period from 2016 to 2021, there was a general increase in the total number of admissions for hepatic failure with and without coma, rising from 3041 in 2016 to 5757 in 2021. The total number of liver transplants performed also showed a gradual increase, from 1437 in 2016 to 1782 in 2021. When examining the race stratification of total admissions for hepatic failure, White individuals consistently represented the largest proportion, ranging from 64.64% to 68.59% across the years. Black individuals accounted for roughly 8.53% to 9.85% of these admissions, while Hispanic individuals comprised a notable and generally increasing percentage, ranging from 14.18% to 18.06%. A similar racial distribution was observed for liver transplantations, with White individuals making up the largest percentage (65.74% to 68.53%), followed by Hispanic individuals (14.18% to 17.36%), and then Black individuals (6.95% to 9.80%).

Table 3, presents the distribution of median household income across national quartiles based on ZIP codes. It shows that the frequency of ZIP codes is relatively similar across all four quartiles. The lowest quartile contains 2279 ZIP codes, representing 24.06% of the total, while the second quartile includes 2320 ZIP codes (24.49%). The third quartile has the highest frequency with 2475 ZIP codes (26.13%), and the highest income quartile contains 2399 ZIP codes, accounting for 25.32% of the total. Cumulatively, the table illustrates how the ZIP codes are distributed across the income spectrum, reaching 100% by the highest quartile, with a total of 9473 ZIP codes analyzed.

Table 4 compares characteristics of all liver transplant recipients with those who died during their hospital stay after the procedure. The mean age was similar in both groups, approximately 51.91 years for all transplants and 52.28 years for those who died, with a non-significant *p*-value of 0.15 from the one-way ANOVA. The distribution of sex (male vs. female) was also not significantly different between the two groups (*p* = 0.24). Similarly, the distribution across different racial groups (White, Black, Hispanic) did not show a significant difference (*p* = 0.23). When examining the median household income quartiles based on ZIP code, there was a slight trend towards a higher number of deaths in lower-income quartiles, but this difference was not statistically significant (*p* = 0.13). Finally, the proportion of patients with a Charlson Comorbidity Index greater than 5 was also similar between the two groups, with around 48.56% in the overall transplant group and 45.83% in the deceased group (*p* = 0.154). Overall, this analysis suggests that the factors examined, including age, sex, race, median household income quartile, and high comorbidity index, were not statistically significant predictors of in-hospital mortality following liver transplantation in this cohort. No significant variance was observed in the age, sex, race, median household income, or percentage of admissions with Charlson comorbidity index > 5 among all liver transplant admissions versus liver transplant admissions that died during the hospital stay.

## 4. Discussion

Our analysis highlights several important findings regarding liver transplantation trends, outcomes, and demographic distributions between 2016 and 2021. Notably, a steady rise in the total number of liver transplants was noted over this period (from 1437 in 2016 to 1782 in 2021), reflecting unwavering efforts to enhance donor availability to facilitate the liver transplant process in the United States. This data mirrors national registry trends. Consequently, liver transplantation demand continues to increase, driven by the growing burden of liver pathologies including non-alcoholic steatohepatitis and alcoholic liver disease [10,11].

One of the noteworthy observations is the relative consistency in mean age at transplant (51–52 years) across all six years. This is generally consistent with reports from large-scale transplant registries, which show that most liver transplant recipients fall in the fifth to sixth decade of life [12,13]. The male predominance (roughly 61–64%) aligns with previous studies suggesting higher rates of end-stage liver disease in men, especially those related to chronic viral hepatitis and alcoholic liver disease [10,14]. This pattern could also be influenced by sex-based differences in healthcare-seeking behaviors and referral rates for transplant evaluation.

Our results also reveal distinct racial distributions among liver transplant recipients, with White patients representing the largest proportion, followed by Hispanic and Black patients. On the other hand, the inpatient mortality for those undergoing transplantation did not differ significantly by race (*p* = 0.23), suggesting that once patients are listed and receive a transplant, short-term survival outcomes may be comparably favorable across racial groups. This finding concurs with other studies that have found no substantial racial disparity in immediate post-transplant survival, even though earlier steps in the transplant evaluation process can be shaped by socioeconomic and cultural factors [15,16]. Interestingly, no significant difference was detected in in-hospital mortality when stratified by insurance payer status or median household income, hinting that, at least in this dataset, once patients reach the point of transplant, short-term mortality outcomes are largely comparable [17,18].

The mean hospital charges and length of stay varied only modestly among the study years. Average total charges ranged around USD 580,000–766,000, and length of stay was typically 20–22 days. These figures are in line with nationwide estimates, given that liver transplantation often involves complex perioperative care, rigorous immunosuppression management, and frequent complications such as infections or rejection episodes [12,19]. Though costs remain substantial, improvements in surgical techniques, perioperative care, and immunosuppressive regimens have collectively contributed to more stable resource utilization patterns over time [20,21].

Another key aspect of our findings is the absence of any statistically significant difference in demographic or clinical factors between patients who survived and those who died in-hospital. Age, sex, race, median household income, and presence of high comorbidity burden (Charlson Comorbidity Index > 5) did not correlate with short-term mortality in this cohort. This result aligns with some studies suggesting that, once on the transplant list, access to consistent multidisciplinary care can mitigate pre-existing disparities [22,23]. Nonetheless, the non-significant *p*-values should be interpreted with caution, as the present analysis focuses on in-hospital mortality alone and may not capture post-discharge complications or longer-term survival patterns [23,24].

Taken together, these findings underscore a continued need for robust, long-term investigations into factors influencing access to transplantation, socioeconomic barriers, and post-transplant survival. While short-term in-hospital mortality appears relatively uniform across demographic groups, ongoing surveillance is warranted to confirm that these patterns hold true beyond the hospitalization index.

This study has several limitations. Its retrospective design using the NIS database makes it vulnerable to coding errors, missing data, and unmeasured confounding variables. The lack of granular clinical information—such as liver disease severity, transplant eligibility criteria, and long-term outcomes—limits the depth of the analysis. Additionally, the cross-sectional nature of the dataset prevents assessment of longitudinal trends or post-transplant survival. Socioeconomic factors, referral biases, and access to care, which likely contribute to the observed disparities, could not be fully accounted for due to limitations in the dataset.

## 5. Conclusions

A nationwide analysis of liver transplantation trends from 2016 to 2021 reveals an overall increase in the number of transplants performed, yet potential disparities exist across racial groups, with White individuals receiving the majority, and by insurance status, where privately insured individuals undergo more transplants compared to those with Medicare or Medicaid. While demographic and clinical factors such as age, sex, and comorbidity index did not significantly predict in-hospital mortality, socioeconomic factors such as median household income quartile show a distribution across transplant recipients, indicating potential access variations. Given the retrospective nature of our study, future research is needed to pinpoint modifiable institutional practices and integrate social determinants of health into transplant evaluations to address these inequities and ensure more equitable access to liver transplantation across all populations.

## Figures and Tables

**Figure 1 medsci-13-00066-f001:**
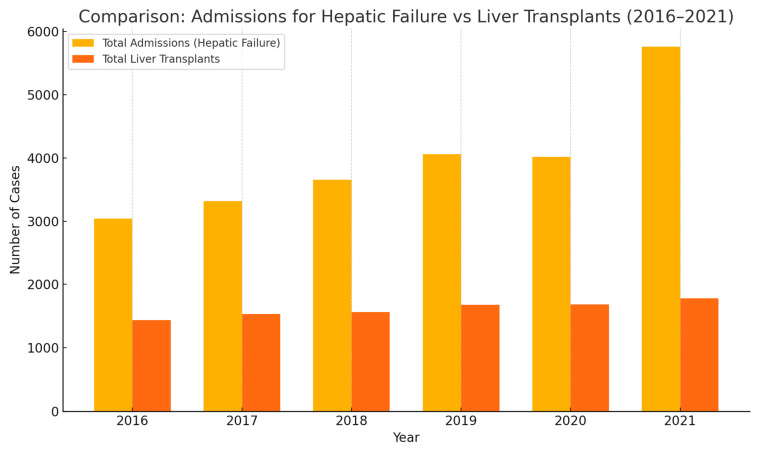
Admissions of hepatic failure from 2016–2021.

**Figure 2 medsci-13-00066-f002:**
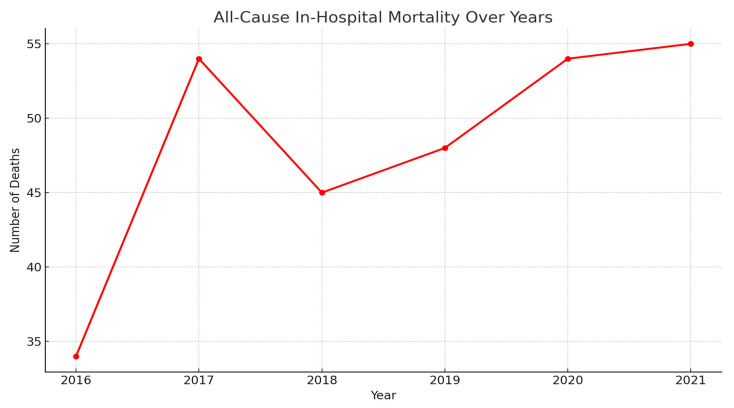
All cause mortality of patients who underwent liver transplant from 2016–2021.

**Figure 3 medsci-13-00066-f003:**
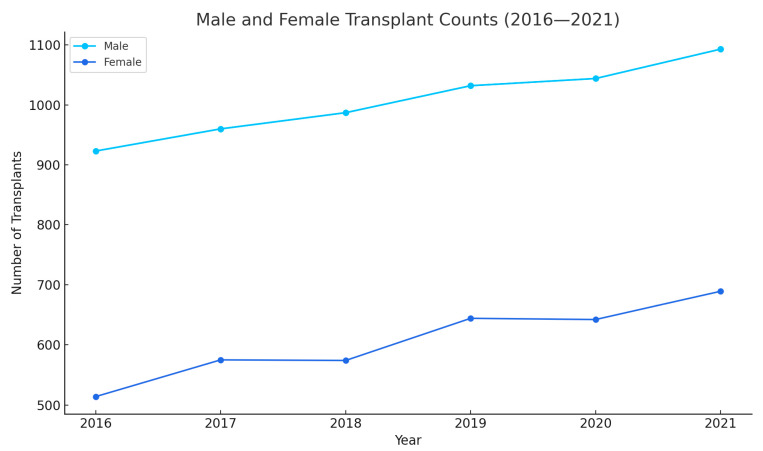
Showing sex distribution over years.

**Figure 4 medsci-13-00066-f004:**
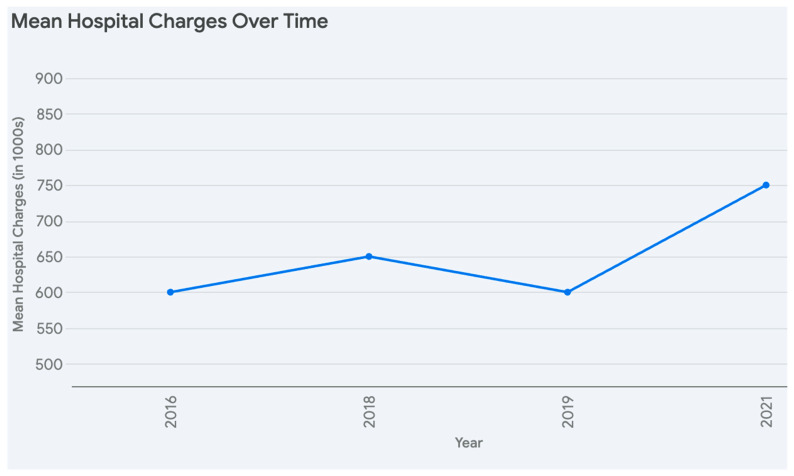
Mean Hospital charges and length of stay from 2016–2021.

**Figure 5 medsci-13-00066-f005:**
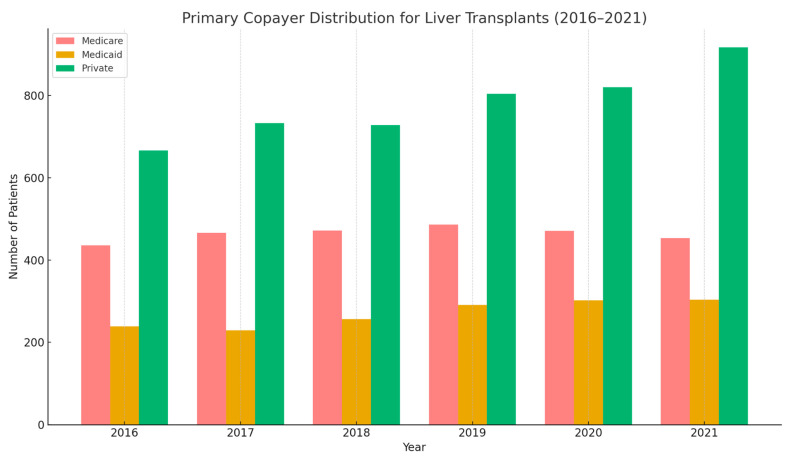
Primary copayer distribution for liver transplants 2016–2021.

**Table 1 medsci-13-00066-t001:** Summary of patients who underwent liver transplant between 2016–2021.

Year	Total Number of Transplants	Mean Age	Sex	Race	All-Cause in Hospital Mortality	Mean of Total Hospital Charges ($)	Mean Length of Stay (Days)	Primary Co Payer
2016	1437	51.80 (50.92–58.68)	Male: 923 (64.23%)	White 866 (68.26%)	34 (2.37%)	578,291.7	21.06 (19.90–22.22)	Medicare 436 (30.64%)
				Black 121 (9.24%)				Medicaid 239 (16.80%)
			Female 514 (35.77%)					
				Hispanic 202 (15.42%)				Private 666 (46.80%)
2017	1535	52.10 (51.25–52.96)	Male 960 (62.54%)	White 946 (65.74%)	54 (3.52%)	584,345.6	20.80 (19.60–22.00)	Medicare 466 (30.56%)
				Black 141 (9.80%)				Medicaid 229 (15.02%)
			Female 575 (37.46)					
				Hispanic 204 (14.18%)				Private 733 (48.07%)
2018	1561	51.86 (51.01–52.71)	Male 987 (63.23%)	White 994 (66.58%)	45 (2.88%)	643,858.8	21.23 (19.86–22.41)	Medicare 472 (30.31%)
				Black 129 (8.64%)				Medicaid 256 (16.44%)
			Female 574 (36.77%)					
				Hispanic 234 (15.67%)				Private 728 (46.76%)
2019	1676	51.98 (51.18–52.78)	Male 1032 (61.58%)	White 1404 (68.53%)	48 (2.86%)	614,708.4	20.47 (19.26–21.68)	Medicare 486 (29%)
				Black 112 (6.95%)				Medicaid 291 (17.36%)
			Female 644 (38.42%)					
				Hispanic 272 (16.88%)				Private 804 (47.97%)
2020	1686	52.12 (51.33–52.91)	Male 1044 (61.92%)	White 1088 (67%)	54 (3.20%)	685,850/4	21.15 (19.94–22.36)	Medicare 471 (27.94%)
				Black 123 (7.57%)				Medicaid 302 (17.91%)
			Female 642 (38.08%)					
				Hispanic 282 (17.36%)				Private 820 (48.64%)
2021	1782	51.59 (50.85–52.33)	Male 1093 (61.34%)	White 1140 (66.55%)	55 (3.09%)	766,002	22.77 (21.45–24.10)	Medicare 453 (25.44%)
				Black 138 (8.06%)				Medicaid 304 (17.07%)
			Female 689 (38.66%)					
				Hispanic 281 (16.40%)				Private 917 (51.49%)

**Table 2 medsci-13-00066-t002:** Race stratification of total admissions with hepatic failure vs. liver transplant.

Year	Total Number of Admissions for Hepatic Failure with and Without Coma	Total Number of Liver Transplants	Race Stratification of Total Admissions for Hepatic Failure with and Without Coma	Race Stratification of Liver Transplantations
2016	3041	1437	White 1978 (68.59%)	White 866 (68.26%)
			Black 284 (9.85%)	Black 121 (9.24%)
			Hispanic 409 (14.18%)	Hispanic 202 (15.42%)
2017	3316	1535	White 2059 (64.99%)	White 946 (65.74%)
			Black 298 (9.41%)	Black 141 (9.80%)
			Hispanic 537 (16.95%)	Hispanic 204 (14.18%)
2018	3653	1561	White 2330 (65.39%)	White 994 (66.58%)
			Black 335 (9.40%)	Black 129 (8.64%)
			Hispanic 623 (17.49%)	Hispanic 234 (15.67%)
2019	4062	1676	White 2671 (67.28%)	White 1404 (68.53%)
			Black 351 (8.84%)	Black 112 (6.95%)
			Hispanic 652 (16.42%)	Hispanic 272 (16.88%)
2020	4019	1686	White 2541 (64.64%)	White 1088 (67%)
			Black 386 (9.82%)	Black 123 (7.57%)
			Hispanic 710 (18.06%)	Hispanic 282 (17.36%)
2021	5757	1782	White 3844 (66.77%)	White 1140 (66.55%)
			Black 491 (8.53%)	Black 138 (8.06%)
			Hispanic 992 (17.23%)	Hispanic 281 (16.40%)

**Table 3 medsci-13-00066-t003:** Income Disparity/Distribution for Liver Transplants (Cumulative).

Quartile	Frequency	Percent (%)	Cumulative (%)
1 (Lowest)	2279	24.06	24.06
2	2320	24.49	48.55
3	2475	26.13	74.68
4 (Highest)	2399	25.32	100
Total	9473	100	

**Table 4 medsci-13-00066-t004:** Factors Affecting All Cause In Hospital Mortality For Liver Transplant Admissions: Cumulative (2016–2021).

	All Liver Transplants	Liver Transplants That Died During Hospital Stay After Procedure	One Way ANOVA (*p*-Value)
**Mean Age**	51.91 (51.57–52.24)	52.28 (50.28–54.27)	0.15
**Sex**	Male: 6039 Female: 3638	Male: 177 Female: 113	0.24
**Race**	White: 6140 Black: 764 Hispanic: 1475	White: 166 Black: 30 Hispanic: 47	0.23
**Median Household Income**	1st Quartile: 2279 2nd Quartile: 2320 3rd Quartile: 2475 4th Quartile: 2399	1st Quartile: 76 2nd Quartile: 72 3rd Quartile: 68 4th Quartile: 64	0.13
**Charlson Comorbidity Index > 5**	812 (48.56%)	22 (45.83%)	0.154

## Data Availability

The data is obtained from the NIS database which is publicly available and contains de-identified patient information.

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
