# Peer review of "Trends and Disparities in Liver Transplantation in the United States: A Nationwide Analysis of Demographic, Clinical, and Socioeconomic Factors (2016–2021)"

_medsci, 2025, doi:10.3390/medsci13020066_

Round 1

Reviewer 1 Report

Comments and Suggestions for Authors

Important and interesting paper. Some specific comments are below:

  1. Liver transplants were identified using ICD-10 PCS codes 0FY00Z1 and 0FY00Z2. Are these validated/right codes? Can authors provide evidence or reference confirming their validating the ICD codes in the diagnosis of liver transplant.

  1. Methods: The study used a national database. Were the analyses weighted to obtain national estimates? Recommend to clearly mention it under Methods section.

  1. Table and figures need to be numbered and need to have clear titles.

  1. There are many tables/figures. Are all of them essential and necessary? For example, are both the table pertaining to median household income and the figure (pie-chart) underneath it necessary? Recommend including only essential tables and figures, and minimizing repetition of the same message.

  1. For the table ‘Mean Age and Sex Distribution Over Years,’ recommend revising the line for mean age to make it more meaningful. Currently the line points toward unmarked area between 0 and 200.

  1. Regarding the figure pertaining to mean charges and length of stay, the line representing length of stay all points toward zero (0). Recommend authors to make it more meaningful.

  1. Regarding the table, “Factors Affecting All Cause In Hospital Mortality For Liver Transplant Admissions,” Are these independent factors following adjusted analysis?
Comments on the Quality of English Language

N/A

Author Response

Liver transplants were identified using ICD-10 PCS codes 0FY00Z1 and 0FY00Z2. Are these validated/right codes? Can authors provide evidence or reference confirming their validating the ICD codes in the diagnosis of liver transplant.

Re: Yes these are valid ICD 10 PCS codes. It can be found online , the author is attaching a link 

https://icd10coded.com/pcs/0FY00Z1/

Methods: The study used a national database. Were the analyses weighted to obtain national estimates? Recommend to clearly mention it under Methods section.

Re: Thank you for this valuable point, however non weighted analyses was done because of the small sample size, added to the method section as well

Table and figures need to be numbered and need to have clear titles.

Re: Yes they were corrected

There are many tables/figures. Are all of them essential and necessary? For example, are both the table pertaining to median household income and the figure (pie-chart) underneath it necessary? Recommend including only essential tables and figures, and minimizing repetition of the same message.

Re: Thank you for this valuable point, 2 of the graphs have been  removed

For the table ‘Mean Age and Sex Distribution Over Years,’ recommend revising the line for mean age to make it more meaningful. Currently the line points toward unmarked area between 0 and 200.

Thank you for this valuable point mean age has been removed and the graph has been changed

Regarding the figure pertaining to mean charges and length of stay, the line representing length of stay all points toward zero (0). Recommend authors to make it more meaningful.

Thank you for this valuable point, the mean length of stay has been removed and the graph has been changed

Regarding the table, “Factors Affecting All Cause In Hospital Mortality For Liver Transplant Admissions,” Are these independent factors following adjusted analysis?

Re: The all cause mortality was an observational data and  no association was mentioned as we used one way anova

a

Reviewer 2 Report

Comments and Suggestions for Authors

This manuscript contains several tables and figures, while with limited text to describe the results/findings, the reviewer can barely understand what's the exact meaning the authors intend to deliver.

Author Response

This manuscript contains several tables and figures, while with limited text to describe the results/findings, the reviewer can barely understand what's the exact meaning the authors intend to deliver.

Thank you for this point, we have reduced the number of images and added text to the results section, changes have been highlighted in red

Reviewer 3 Report

Comments and Suggestions for Authors

The authors conducted a study titled "Trends and Disparities in Liver Transplantation in the United States: A Nationwide Analysis of Demographic, Clinical, and Socioeconomic Factors (2016–2021)."

While I do not have a detailed understanding of the NIS dataset, I believe the limitations of the variables presented in this study make it difficult to draw definitive conclusions. Furthermore, given the absence of various clinical factors, it is challenging to support the authors’ conclusions based solely on this cross-sectional data analysis. Rather than attempting to draw inferential conclusions through statistical analysis, it could be more appropriate to present the findings as a descriptive report highlighting observed trends.

Author Response

While I do not have a detailed understanding of the NIS dataset, I believe the limitations of the variables presented in this study make it difficult to draw definitive conclusions. Furthermore, given the absence of various clinical factors, it is challenging to support the authors’ conclusions based solely on this cross-sectional data analysis. Rather than attempting to draw inferential conclusions through statistical analysis, it could be more appropriate to present the findings as a descriptive report highlighting observed trends.

Reply: Thank you for this valuable point, I have changed the conclusion to being more descriptive, the all cause mortality is an observational data and we only used one way anova

Round 2

Reviewer 2 Report

Comments and Suggestions for Authors

This manuscript discusses the topic of trends and disparities of liver transplantation in the United States, including the demographic, clinical, and socioeconomic factors. While the concept is a valuable goal, the study suffers from notable methodological which limit its significance. The topic is not novel, and the manuscript reiterates previously published findings without offering significant new insights or evidence in this field.

Reviewer 3 Report

Comments and Suggestions for Authors

The revision appears to have appropriately addressed the concerns regarding the dataset and statistical analysis, and the interpretation has been suitably modified.

Congratulations on the improvement.